# Estimating Pore Water Electrical Conductivity of Sandy Soil from Time Domain Reflectometry Records Using a Time-Varying Dynamic Linear Model

**DOI:** 10.3390/s18124403

**Published:** 2018-12-13

**Authors:** Basem Aljoumani, Jose A. Sanchez-Espigares, Gerd Wessolek

**Affiliations:** 1Department of Ecology, Ecohydrology and Landscape Evaluation, Technische Universität Berlin Ernst-Reuter Platz 1, 10587 Berlin, Germany; 2Department of Statistical and Operational Research, Universitat Politècnica de Catalunya (UPC), Jordi Girona, 31, 08034 Barcelona, Spain; josep.a.sanchez@upc.edu; 3Department of Ecology, Technische Universität Berlin Ernst-Reuter Platz 1, 10587 Berlin, Germany; gerd.wessolek@tu-berlin.de

**Keywords:** electrical conductivity, relative dielectric permittivity, time domain reflectometry, kalman filter, dynamic linear model

## Abstract

Despite the importance of computing soil pore water electrical conductivity (*σ_p_*) from soil bulk electrical conductivity (*σ_b_*) in ecological and hydrological applications, a good method of doing so remains elusive. The Hilhorst concept offers a theoretical model describing a linear relationship between *σ_b_*, and relative dielectric permittivity (*ε_b_*) in moist soil. The reciprocal of pore water electrical conductivity (1/*σ_p_*) appears as a slope of the Hilhorst model and the ordinary least squares (OLS) of this linear relationship yields a single estimate ($\hat{1/\sigma_{p}}$) of the regression parameter vector (*σ_p_*) for the entire data. This study was carried out on a sandy soil under laboratory conditions. We used a time-varying dynamic linear model (DLM) and the Kalman filter (Kf) to estimate the evolution of *σ_p_* over time. A time series of the relative dielectric permittivity (*ε_b_*) and *σ_b_* of the soil were measured using time domain reflectometry (TDR) at different depths in a soil column to transform the deterministic Hilhorst model into a stochastic model and evaluate the linear relationship between *ε_b_* and *σ_b_* in order to capture deterministic changes to (1/*σ_p_*). Applying the Hilhorst model, strong positive autocorrelations between the residuals could be found. By using and modifying them to DLM, the observed and modeled data of *ε_b_* obtain a much better match and the estimated evolution of *σ_p_* converged to its true value. Moreover, the offset of this linear relation varies for each soil depth.

## 1. Introduction

Salinization reduces crop productivity, decreases profitability, and causes land scarcity [1]. Thus, it decreases the world’s agricultural productivity and causes a global income loss of US$ 12 billion per year [2]. Extracting soil solution by suction or using saturated paste conductivity measurements are the common methods to determine the electrical conductivity of soil pore water (*σ_p_*) as an indicator of the soil salinity; however, they are labour- and cost intensive. There is no evidence that all ions are collected in the sample extract [3]. For soil salinity assessment, it is important to look for practical methods that evaluate the soil salinity state temporally and spatially. These methods help to correctly evaluate soil salinity evolution and reasonably predict its values [4,5,6,7,8,9]. In recent times, soil electromagnetic sensors have been used to estimate bulk electrical conductivity (*σ_b_*). Then, methods are required to transform *σ_b_* to *σ_p_* [3,6,10].

According to Wyllie and Southwick [11], three conductance pathways, see Figure 1, contribute to the *σ_b_* of a soil: (i) solid phase pathway through soil particles that are continuous contact with one another, (ii) liquid phase pathway through dissolved ions in the soil water inhabiting the large pores, and (iii) a liquid–solid interphase pathway through exchangeable cations like surfaces of clay minerals. Electrical conductivity (EC) in the liquid phase (*σ_p_*) is used to estimate the soil salinity, a high EC refers to a high concentration of soluble salts, and vice versa. The *σ_p_* could be estimated if the relationship between *σ_p_*, *σ_b_*, and water content (*θ*) is fixed [12,13,14]. The discovered linear correlation between the soil relative dielectric permittivity (*ε_b_*) and *σ_b_* values [15] enabled Hilhorst [3] to convert *σ_b_* to *σ_p_* by using a theoretical model. According to Hilhorst, *σ_p_* can be determined from the equation:(1)$$\sigma_{p}=\frac{\varepsilon_{p}\sigma_{b}}{\varepsilon_{b}-\varepsilon_{\sigma_{b}=0}}$$
where *σ_p_* is the pore water electrical conductivity (dS/m); *ε_p_* is the relative dielectric permittivity of the soil pore water (dimensionless), *ε_b_* is the relative dielectric permittivity of the bulk soil (dimensionless, relative dielectric permittivity is dimensionless since it is a ratio of permittivity of medium to the permittivity of free space), *σ_b_* is the bulk electrical conductivity (dS/m), *ε_σb=0_* is the relative dielectric permittivity of the soil when the bulk electrical conductivity is 0 (dimensionless). However, *ε_σb=0_* appears as an offset of the linear relationship between *ε_b_* and *σ_b_*. The Hilhorst model [3] concluded that his method could be validated for water contents between 0.10 and saturation and for a conductivity of the pore water up to 0.3 S m^−1^. He found that *ε_σb=0_* depends on soil type and varies between 1.9 and 7.6. He recommended using 4.1 as a generic offset. Many studies applied the deterministic Hilhorst model [3] in their experiments to convert *σ_b_* into *σ_p_* but they did not use the same offset to achieve their study objective. For example, some studies concluded their work by using different offsets (within the range of 3.67 to 6.38) according to the soil type [10]. The producer of capacitance soil moisture sensors 5TE [16] recommends the use of an offset *ε_σb=0_* of 6 while another study found that an offset *ε_σb=0_* = 6 does not present a good linear relationship between *ε_b_* and *σ_b_* [17]. The WET sensor (Delta-T Device Ltd., Cambridge, UK) is a frequency domain dielectric sensor. It has been designed to estimate the *σ_p_* based on the Hilhorst model [3] and incorporate the standard offset *ε_σb=0_* = 4.1 of the model in the software of the device. By applying the Hilhorst model [3] in a saline gypsum-influenced soil, the accuracy of the WET sensor in predicting *σ_p_* was very poor when using the offset model =4.1 [18]. Another study used a WET sensor for experimental measurements in the laboratory using four different soils (sand, sandy loam, loam, and clay) [9] and found that the offset depends on both soil type and *σ_p_*, where it becomes larger for larger *σ_p_*. Moreover, oscillator frequency and sensor circuitry could affect the estimation of *ε_b_* and water content (θ) [19].

There are three elementary causes responsible for why the deterministic system and control theories do not produce a totally sufficient means of performing this analysis and design: (i)many effects are left unknown since the objective of the model is to represent the main modes of system response,(ii)deterministic models are driven not by only our own control inputs but also disturbances which we can neither control nor model deterministically, and(iii)sensors do not offer exact readings of chosen quantities but present their own system dynamics and distortions as well and these devices are noise corrupted [20]. Despite the importance of computing *σ_p_* from *σ_b_*, a good method for doing so remains elusive (Campbell [16], personal communication).

Solute transport and water flow in the unsaturated zone are normally derived from the classical Richards equation and the convection–dispersion equations. Deterministic explanations of these equations are important aspects of research; due to soil heterogeneity at a variety of spatial scales, these equations for predicting actual field-scale processes are being increasingly questioned [21]. Therefore, some researchers working on soil heterogeneity concluded that for the evolution of soil water and solutes, it is more desirable to use stochastic models rather than constant values, where the parameters of stochastic transport models are treated as random variables with discrete values assigned according to a given probability distribution [21,22,23,24,25,26]. Among stochastic models, many studies used Kalman filtering in hydrological applications. A Kalman filter is an optimal recursive data processing algorithm that recursively couples the most recent measurements into the linear model to update the model state output [27]. Under the assumption that the linear system is a stochastic process with Gaussian noises, it produces the best estimation with minimum mean square error and it has been widely used in hydrological models to optimally merge information from the model simulations and the independent observations with appropriate modeling [28,29,30,31,32].

In previous work, we installed frequency domain reflectometry (FDR) sensors (5 TE), which are commercially available from METER Group, Inc. USA, in field conditions at different depths where the soil is heterogeneous to estimate *σ_p_* [22]. We used ε*_b_* and *σ_b_* observations to modify the Hilhorst deterministic model [3] to a stochastic model using a time-varying dynamic linear model and Klaman filter before studying the linear relationship between them.

In this study, we used Time Domain Reflectometry (TDR) sensors (FP/mts), which are commercially available from Easy Test, Poland, to measure *ε_b_* and *σ_b_* in laboratory conditions where the soil is homogeneous. Then, we tried to use the Hilhorst model [3] to convert *σ_b_* to *σ_p_*. Later, we could show the weakness of applying the deterministic Hilhorst model [3] even in homogeneous soils. Thus, we are aiming to adapt this approach to a stochastic model under laboratory conditions. Thus, we used one homogeneous soil type to accurately estimate the changes in *σ_p_* over time and to conclude whether the model offset is constant or if it changes in one soil profile.

## 2. Material and Methods

### 2.1. The Column Experiment

To achieve the objective of this study, we used two soil columns with a height of 55 cm provided by a sprinkler, see Figure 2. The lower boundary was controlled using a vacuum pump at a constant pressure head of −30 hPa. The columns were packed with a density of 1.4 g/cm^3^. The substrate was sand, 80% of which was fine sand. The water content during packing was approximately 4 m^3^/m^3^. The TDR and soil temperatures sensors were installed in four depths: 7, 21, 35, and 48 cm. Since the soil is sand, the soil relative dielectric permittivity (*ε_b_*), bulk electrical conductivity (*σ_b_*), and temperature were measured every 5 min to obtain enough observations for modeling. 

Additionally, porous suction cups for taking soil solution samples were installed at each depth to validate the results of our model. The lower boundary of the column uses a membrane to let the water drain. Drainage water was collected in a bottle under −30 hPa vacuum, which is supplied in the range from −20 to −30 hPa. The sprinkler is 5 cm above the soil surface and allows water to drop through small nozzles. Five irrigation events using KCl solution with different electrical conductivities were applied. The first three events were irrigated with 20 dS/m of KCl, then the fourth, and fifth events with 30 dS/m of KCl. The flux was approximately 1 l/h. The columns were free of salt at the beginning and before the irrigations events started. The TDR probes are FP/mts commercially available from Easy Test, Poland, and have been calibrated in air and deionized water. The temperature probes are Thermistors of the type 2k252 (type Fenwal UUA 32J3) with a range of −20 up to 60 °C. Soil temperature data (*T**soil*) were used to estimate the relative dielectric permittivity of the soil pore water directly (*ε_p_*): (2)$$\varepsilon_{p}=80.3-0.37\left(T_{soil}-20\right)$$

To apply the dynamic linear model and the Kalman filter, a time series of the variable of interest is needed [27]. In our study, time series of *ε_b_*, *σ_b_*, and *ε_p_* are required to estimate *σ_p_*. Therefore, we used five irrigation events with two levels of KCL solution to obtain the variation of these variables over time for each depth. In total, 289 observations were made of *σ_b_*, *ε_p_*, and *ε_b_* for each soil depth and these were used to estimate both the offset *ε_σb=0_* of the modified Hilhorst model [3] and the evolution of *σ_p_* at its corresponding depth, of which 144 observations were used to validate their forecasts.

### 2.2. Time-Varying Dynamic Linear Model

In general, the state space model is identified by two assumptions, (i) there is a hidden or latent process *x_t_* called the state process. The state process is assumed to be a Markov process, where past and future values of *x_t_* are independent, conditional on the present *x_t_*, ({*x_s_*, *S* > *t*}, and {*x_s_*, *S*< *t*} are independent on the *x_t_*), (ii) the observations, *y_t_* are independent given the states *x_t_*. This means that the dependence among the observations is generated by states. The dynamic linear model (DLM) or linear Gaussian state space model, in its simple form, employs a first-order, p-dimensional vector autoregression as the state equation:(3)$$x_{t}=x_{t-1}+w_{t}\;w_{t}\sim N\left(0,W_{t}\right)$$

We do not observe the state vector *x_t_* directly, but only a linear transformed version of them with noise added, say:(4)$$y_{t}=A_{t}x_{t}+v_{t}\;v_{t}\sim N\left(0,V_{t}\right)$$
*y_t_* is an m-dimensional vector, representing the observation at time *t*, $A_{t}$ is a *q × p* measurement or observation matrix. Equation (4) is called the observation equation, in which $v_{t}$, $w_{t}$ are the Gaussian white-noise errors. The evolution variances are $V_{t}$, $W_{t}$ and can be estimated from available data using maximum likelihood or Bayesian techniques.

In this study, we modified the deterministic Hihlorst model (1) to a stochastic one. The model (1) has the variables *σ_b_*, *ε_p_*, σ*_p_*, *ε_b_*, and ε_σ*b = 0*_. The *σ_p_* and *ε_σb = 0_* are unobserved and they need to be estimated by the state Equation (3) as *x_t_*, while *σ_b_*, *ε_p_*, and *ε_b_* are observed by the sensors (*ε_p_* is calculated from Equation (2) using soil temperature sensor data) and represented by observation Equation (4) as *y_t_*.

The R [33] package, dlm [34], provides an integrated environment for Bayesian inference using DLM, and the package includes functions for Kalman filtering and smoothing, as well as for maximum likelihood estimation.

## 3. Results and Discussion

### 3.1. Deterministic Model

The offset of the Hilhorst model [3] can be calculated from Equation (1): (5)$$\varepsilon_{b}=\left(1/\sigma_{p}\right)\varepsilon_{p}\times \sigma_{b}+\varepsilon_{\sigma_{b}=0}$$

We derived the offset (ε_σ*b = 0*_) from this linear model after using measurements of *ε_b_* and *σ_b_*. For example, applying the ordinary least squares (OLS) on measurements of *ε_b_* and *σ_b_* obtained from soil column 2 data during the third irrigation at a depth of 21 cm, Table 1 shows that the offset of the linear relationship between *ε_b_*–*σ_b_* is 9.41. Further, the single estimate of the slope ($\hat{1/\sigma_{p}}$) of the regression parameter vector (1/*σ_p_*) for the entire data set is very small. Thus, the estimated soil pore water electrical conductivity (*σ_p_* ) is too high compared with the EC meter value, see Table 2. Afterward, we applied the Durbin–Watson test in order to test if there was any autocorrelation between the residuals of the regression. Table 3 shows that there is an extremely strong and positive autocorrelation, meaning that the result of that regression is not valid. 

For each irrigation event, we obtained one solution sample at each depth by using porous suction cups. Unfortunately, some samples did not have enough solution to measure their electrical conductivity using the EC meter device. Table 2 shows the values of EC measured by the EC meter device. The table shows eight EC values from the EC meter according to the depth and irrigation event number for each soil column. Due to the variability in the water flow in unsaturated soil, we observed in our experiment a variation in the time needed to collect the solution sample. More time was required to collect enough solution for the EC meter device when a greater number of ions gathered in the sample resulting in a high EC value of the sample. Therefore, there is a difference in the EC values between the soil columns at the same depth, see Table 2. We applied a modified Hilhorst model on the eight time-series data corresponding to Table 2 (depth, irrigation event, and soil column) to compare our finding of *σ_p_* obtained from our modified model to the values of *σ_p_* obtained by the EC meter device, see Table 2.

The reason for choosing 1 l/h for the irrigation rate and 5 min for the irrigation interval is visualized in Figure 3. At each depth, we could see how the bulk electrical conductivity responds to the irrigation event.

### 3.2. Time-Varying Linear Dynamic Model (LDM)

The deterministic Equation (5) can be modified into the time-varying DLM for observation and unobservable (state) models. In this case, the observation data are the soil relative dielectric permittivity (*ε_b_*), bulk electrical conductivity (*σ_b_*), and the relative dielectric permittivity (*ε_p_*), while the unobservable data are the offset (*ε_σb=0_*) and pore water electrical conductivity (*σ_p_*). Equation (4) can be modified to the time-varying DLM as follows:The observation equation can be obtained by modifying the Hilhorst model [3] (written in Equation (5)) into a stochastic equation, in accordance with Equation (4) as follows:
(6)$${\left(\varepsilon_{b}\right)}_{t}={\left(\varepsilon_{\sigma_{b}}=0\right)}_{t}+{\left(\varepsilon_{p}*\sigma_{b}\right)}_{t}{\left(\frac{1}{\sigma_{p}}\right)}_{t}+v_{t}\;v_{t}\sim N\left(0,\sigma_{v}{}^{2}\right)$$The state equation (unobservable data) in Equation (3) is *ε_σb = 0_*, and the slope, 1/*σ_p_*. They can be converted to the unobservable state equation of the time-varying DLM according to Equation (3). The unobservable state equation can be arranged as follows:
(7)$$\{\begin{array}{c} {\left(\varepsilon_{\sigma_{b}}=0\right)}_{t}={\left(\varepsilon_{\sigma_{b}}=0\right)}_{t-1}\\ {\left(\frac{1}{\sigma_{p}}\right)}_{t}={\left(\frac{1}{\sigma_{p}}\right)}_{t-1}+w_{t}w_{t}\sim N\left(0,{\left(\sigma_{w}\right)}_{t}^{2}\right) \end{array}$$

Here, we consider *ε_σb = 0_* as a constant. The actual value is related only to its past value. The slope 1/*σ_p_* changes over time and its actual value is related to its past value plus the Gaussian white-noise errors ($w_{t}$). We applied the equation in reverse order to estimate the state variables (*ε_σb = 0_* and *σ_p_*) at all time points from a complete series of the soil relative dielectric permittivity (*ε_b_*). This process is known as smoothing.

An example of the evolution of *ε_b_*, *ε_p_*, and *σ_b_* data needed for the Hilhorst model [3] is shown in Figure 4. By applying the Equations (6) and (7) using DLM and the Kalman filter on the eight time-series data, we see in Figure 5 the observed and predicted time series of the soil relative dielectric permittivity (*ε_b_*). The predicted and observed values of *ε_b_* agree reasonably well. The mean absolute prediction error (MAPE) for the time series never exceeded 0.02.

Since the prediction of the soil relative dielectric permittivity (*ε_b_*) is valid, the estimation of the electrical conductivity of pore water (*σ_p_*) and the offset *ε_σb = 0_*, see Equation (7), are also valid because they are used in the prediction of the soil relative dielectric permittivity (*ε_b_*) and have converged to their true values. The evolution of *σ_p_* over time obtained by DLM is presented in Figure 6; it shows the importance of using DLM because it obtained all the changes of *σ_p_* over time and not a single value of *σ_p_* for the entire data set. Another interesting aspect is that Figure 6 shows the changes in the model offset for each irrigation event at each depth. This finding is very important since it shows that the offset does not depend on the soil type [10,14,15,16] nor on the soil type and salinity [17] when two columns with the same type soil are used, as in this study. Moreover, in Figure 6 we put the corresponding value of *σ_p_* measured by the EC meter device for each depth according the irrigation event and soil column number.

Comparing the mean evolution of *σ_p_* values obtained from our modified Hilhorst model, see Figure 6, with the single corresponding EC value obtained from porous suction cups and measured by the EC meter device, see Table 2, we found that they agree very well (*R*^2^ = 72%).

From these results, three advantages are evident when using DLM and a Kalman filter to estimate *σ_p_* in two homogenous soil columns; first, we observed that the offset value of the Hilhorst model does not depend on the soil type and *σ_p_* and it changes in the same soil profile. Secondly, we obtained the changes in the estimated *σ_p_* over time and not just a single value as a coefficient for the entire data set. Third, the estimated changes in *σ_p_* occur instantly and save time and labor costs.

## 4. Conclusions

In this study, we applied the *ε_b_*-*σ_b_* linear relationship to homogeneous soil column data obtained from TDR sensors. We found an extremely strong positive autocorrelation between the residuals of the regression analysis. When residuals are correlated, the least squares method is not the most efficient model coefficient estimator. By modifying the regression by a time-varying dynamic linear model (DLM), the match between the observed and modeled data of *ε_b_* is significantly improved and the estimated evolution of *σ_p_* converges to its true value. Moreover, in this study, we used two homogeneous soil columns with the same condition to show that the offset of the Hilhorst model [3] is not constant, as suggested for all moist soil or, as others suggested, that it is soil-type-dependent [10,14,15,16] or soil-type- and salinity-dependent [17]. We repeated the experiment to show that the offset changes even in the same soil type and the same conditions. A dynamic linear model enables the capture of the offset changes and it shows the importance of calculating it simultaneously when estimating *σ_p_* using the Hilhorst model. The next promising step would be programming and inserting these models into the TDR software in order to estimate the soil pore water electrical conductivity (*σ_p_*) from senor records directly.

## Figures and Tables

**Figure 1 sensors-18-04403-f001:**
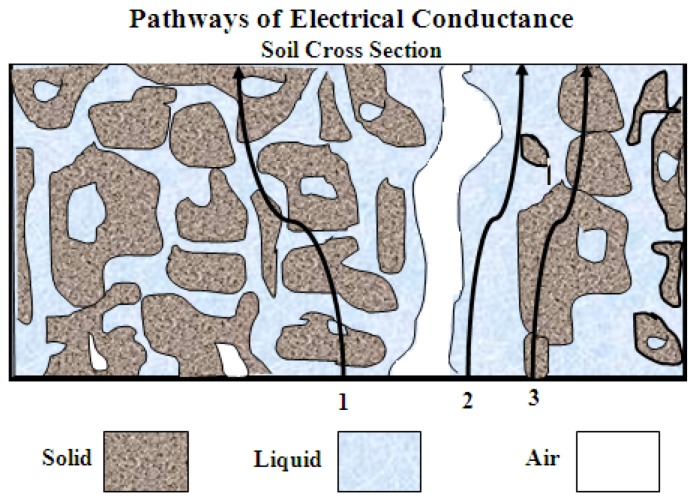
Three conductance pathways for the *σ_b_* measurements, inspired by Wyllie and Southwick [11].

**Figure 2 sensors-18-04403-f002:**
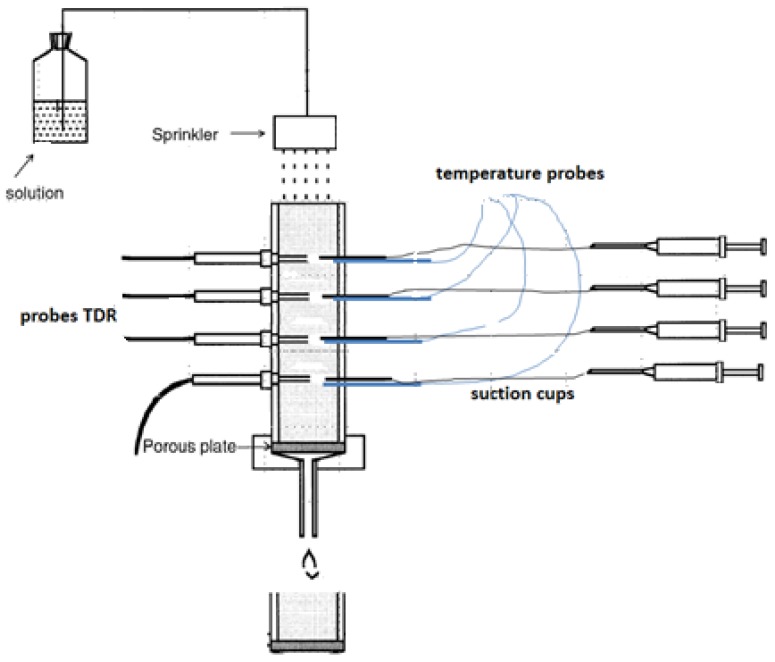
Measurement set.

**Figure 3 sensors-18-04403-f003:**
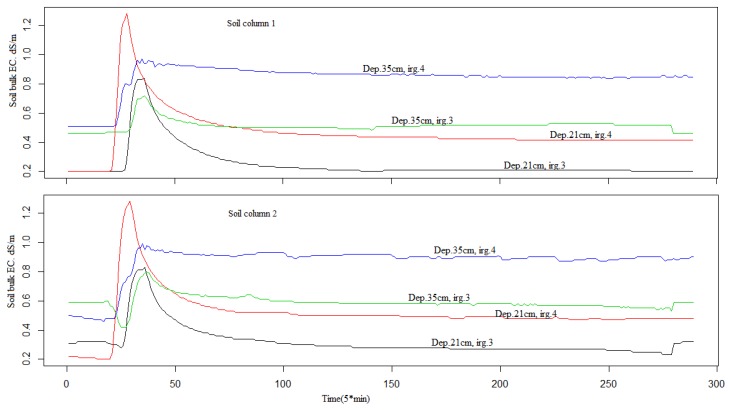
Bulk electrical conductivity (*σ_b_*) in the two soil columns for two irrigation events (N°.3 and N° 4) at two depths (21 cm and 35 cm). Series peaks are related to time irrigation.

**Figure 4 sensors-18-04403-f004:**
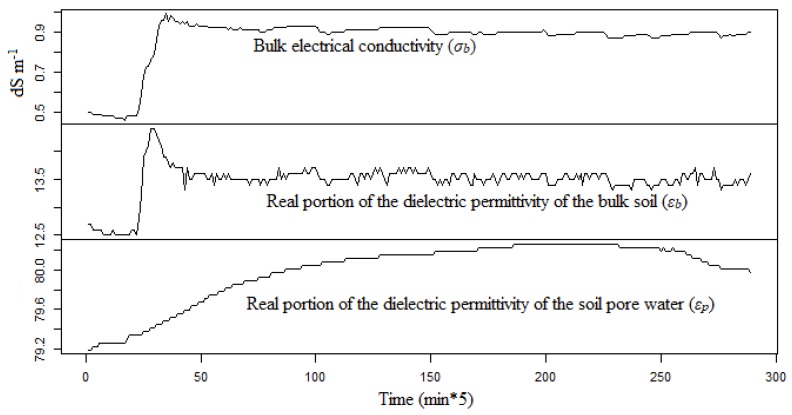
Known variables for the Hilhorst model (*σ_b_*, *ε_b_*, and *ε_p_*); data from soil column 2, depth 35 cm and irrigation event N°4.

**Figure 5 sensors-18-04403-f005:**
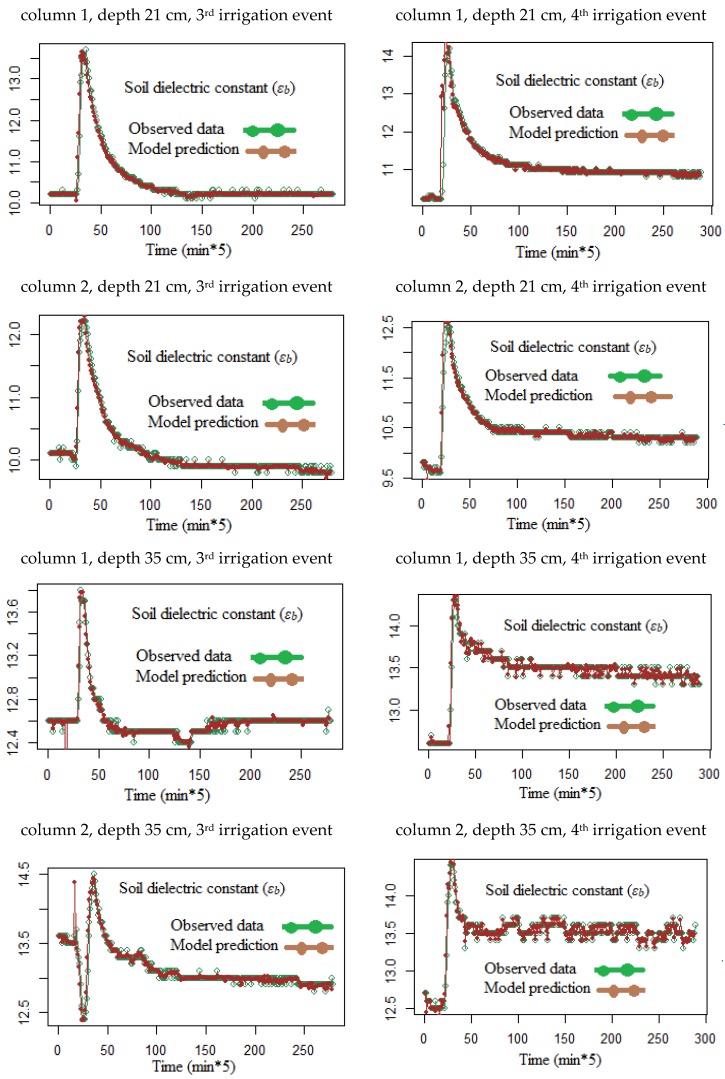
Observed and predicted soil relative dielectric permittivities according to the soil column number, depth, and irrigation event.

**Figure 6 sensors-18-04403-f006:**
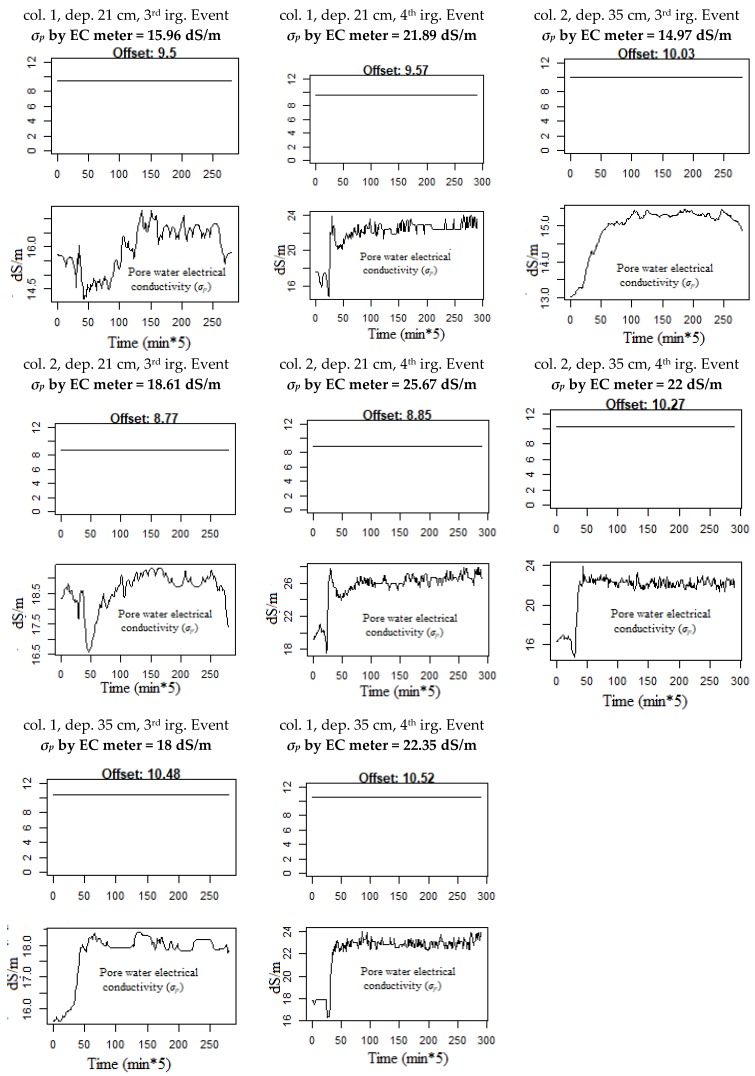
Estimation of the unobservable data (*ε_σb = 0_* and *σ_p_*) by applying the time-varying dynamic linear model (DLM) and the Kalman filter on the data according to the soil column number (col.), depth (dep.), and irrigation event (irg. Event), the corresponding of *σ_p_* by EC meter device is given for each estimated *σ_p_*.

**Table 1 sensors-18-04403-t001:** Estimated parameters gained from the linear regression analysis.

	Estimate	Std. Error	*t* Value	Pr (>|t|)
*ε_σb = 0_*	9.411	8.591 × 10^‒3^	1095.4	<2 × 10^‒16^ ***
1/*σ_p_*	6.963 × 10^‒^^4^	4.461 × 10^‒6^	156.1	<2 × 10^‒16^ ***

Significance: *** *p* < 0.001.

**Table 2 sensors-18-04403-t002:** Electrical conductivity of the soil solution (dS/m) according to soil column number, irrigation event, and depth (cm); it is collected by porous suction cups and measured by an electrical conductivity (EC) meter device.

Soil Column 1				Soil Column 2			
Irrigation Event 3		Irrigation Event 4		Irrigation Event 3		Irrigation Event 4	
Depth: 21 cm	Depth: 35 cm	Depth: 21 cm	Depth: 35 cm	Depth: 21 cm	Depth: 35 cm	Depth: 21 cm	Depth: 35 cm
15.96	18	21.89	22.35	18.61	14.97	25.67	22

**Table 3 sensors-18-04403-t003:** Durbin–Watson test for linear regression *ε_b_-σ_b_*.

Lag	Autocorrelation	D-W Statistic	*p*-Value
1	0.852	0.278	0

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
