# Peer review of "Estimating Pore Water Electrical Conductivity of Sandy Soil from Time Domain Reflectometry Records Using a Time-Varying Dynamic Linear Model"

_sensors, 2018, doi:10.3390/s18124403_

Reviewer 1 Report

Review Sensors-391425

General comment

The authors, having TDR-measured values of soil relative dielectric constant εb, soil bulk electrical conductivity σb and soil temperature, (which is related to the water relative dielectric constant εp) at different depths in two uniformly packed sandy soil columns for a certain time duration, (measurements taken every 5 minutes), transformed the deterministic model of Hilhorst (εb=(1/ σp)* σbσb=0) into a stochastic one in order to capture deterministic changes of σp-1, σp denoting the soil pore water electrical conductivity. This approach, may lead to some “practical methods which could evaluate soil salinity state temporally and spatially”. It appears an ambitious endeavor , although the experiment  was conducted in the laboratory in packed columns of sand. Having said this, I would not decline the paper, but to my opinion moderate to major revision is necessary for an improved version, which would be more easily comprehensible  to the readers of “Sensors”.

More details

Title

The title should include soil pore water electrical conductivity and not just electrical conductivity.

Abstract

l.23 Kalman

Introduction

L35-36 Perhaps these two lines could be clearer. The decrease of agricultural productivity, as it is given, does it apply world-wide or in certain regions or countries?

Figure 1 and the text below (lines 46-49). It  seems o.k. and gives a vision of the various conductance pathways. It is, though, more appropriate to cite the scientists who addressed this issue first. (Sauer et al. Industrial and Engineering Chemistry, 1955, pp.2187-2193). Of course you can also refer to [10] who made it, perhaps, clearer and more popular.

l.56 and elsewhere. The expression “unit-less”, perhaps is better if it is replaced by the adjective dimensionless, and at the same time accompanied by the explanation, as for example with the case of dielectric permittivity, which certainly possesses dimensions in either metric systems (CGS or MKSA, or…), and its more convenient expression as relative dielectric permittivity, being the ratio with the dielectric constant of the vacuum as denominator.

l. 61 I suggest being consistent with the units. In all figures, tables and the text you are using dS/m and not S.m-1  or  S/m. I understand that these units were used by  [3].

l. 62 generic or general?

l. 65 Apart from [15-17], perhaps another quite relevant work could be cited.( G. Kargas and P. Kerkides, 2010. Evaluation of a dielectric sensor for measurement of soil water electrical conductivity. Journal of Irrigation and Drainage Engineering (ASCE) 136 (8):553-558).

l. 66 It  is circuitry not circuity.  Also,  apart from [8] the work of Kargas, George; Persson, Magnus; Kanelis, George; Markopoulou, Ioanna; Kerkides, Petros. Prediction of soil solution electrical conductivity by the permittivity corrected linear model using a dielectric sensor.  Journal of Irrigation and Drainage Engineering, Vol. 143, No. 8, 04017030, 01.08.2017 is quite relevant.

l. 87 heterogeneous and not heterogeneity.  Also, to modify and not modified…

Materials and methods + Results and discussion

This section, to my opinion needs some further information and a more detailed description of the experiment and how this was performed. (Being an extension of a work presented in a Conference, this could be considered a good chance for the authors to feel free to develop and present their contribution without restrictions). For example, why did they use two columns and not more, or just one. Is this an insinuation of spatial variability? Why the chosen flux (1L/h) and what was the hydraulic conductivity at saturation Ks of the column. What is the purpose of the irrigation events, since the salinity level or the electrical conductivity of the moistening solution, which is the dominant factor, had only two different values, (20 and 30dS/m), and in any case, how these values were compared with EC measured independently with the EC meter device, through the suction cups, or the values predicted by the Hilhorst model. Of course, in tables 2 and 3, as well as in fig.3 there is some information on the above. More explanation, on the findings, why for example, there are differences between the two columns, when these are moistened with equal EC KCl-solution and measurements refer to the same depth? The authors say something, concerning the feasibility of getting enough soil solution through the suction cups, but they do not report the number of these measured values and their comparison with the estimated ones. In fig. 6, I would expect to see how these findings do compare with the measured σp values. Perhaps, some information from table 2 could be provided OR some more explanatory comments. On the other hand, fig. 1 showing the conductance pathways seems not to be further commented in the text, thus, it appears to be redundant. The Gaussian white-noise errors wt and vt should be given below the lines of their associative equations and the symbols associated with the standard deviation of the normal probability distribution they are assumed to follow must not be given with the same symbol wt. It is confusing.

In table 2 where  σp values are shown, are these values the mean of a number of measured values or what?

l. 98 What do you mean by the words “a two soil columns”

l. 99 What do you mean by the words “by an irrigation head at …”

l.139. Hilhorst

l. 192. The word “third” contradicts the legend of fig.4

Author Response

Dear Editor and Referees,

Thank you for your very careful review of our manuscript, and for the comments, corrections that ensued. A major revision of the manuscript has been carried out to take all of them into account. this this cover letter, we believe that the manuscript has been significantly improved.

In the present letter, we detail the major changes in the manuscript to address the issues raised by the review

#Response to Reviewer 1 Comments

General comment

The authors, having TDR-measured values of soil relative dielectric constant εb, soil bulk electrical conductivity σb and soil temperature, (which is related to the water relative dielectric constant εp) at different depths in two uniformly packed sandy soil columns for a certain time duration, (measurements taken every 5 minutes), transformed the deterministic model of Hilhorst (εb=(1/ σp)* σbσb=0) into a stochastic one in order to capture deterministic changes of σp-1, σp denoting the soil pore water electrical conductivity. This approach, may lead to some “practical methods which could evaluate soil salinity state temporally and spatially”. It appears an ambitious endeavor,

Thank you for your comments

although the experiment was conducted in the laboratory in packed columns of sand. Having said this, I would not decline the paper, but to my opinion moderate to major revision is necessary for an improved version, which would be more easily comprehensible to the readers of “Sensors”.

More details

Point 1: Title

The title should include soil pore water electrical conductivity and not just electrical conductivity.

Response 1: Thank you for your comment. in our work, estimation the pore water EC comes from evaluating the relationship between bulk EC and soil relative dielectric constant, we chose our title inspired from other studies work in this topic (i.e., Persson 2002), actually before we got the reviewer’s report, I wrote to the editor for the possibility to change the title and let it describes better our objective, taking your comment into account, we change the title in the revised manuscript to:

“Estimating pore water electrical conductivity of sandy soil from time domain reflectometry records using time-varying dynamic linear model” Line 2-3

Abstract

Point 2: l.23 Kalman

Response 2: we correct it, thank you.

Point 3: Introduction

L35-36 Perhaps these two lines could be clearer. The decrease of agricultural productivity, as it is given, does it apply world-wide or in certain regions or countries?

Response 3: we modified it. In the revised manuscript, L33-34

Point 4: Figure 1 and the text below (lines 46-49). It seems o.k. and gives a vision of the various conductance pathways. It is, though, more appropriate to cite the scientists who addressed this issue first. (Sauer et al. Industrial and Engineering Chemistry, 1955, pp.2187-2193). Of course you can also refer to [10] who made it, perhaps, clearer and more popular.

Response 4: we agree, in the revised manuscript we put the reference in the figure 1 Line 44

Thank you

Point 5: l.56 and elsewhere. The expression “unit-less”, perhaps is better if it is replaced by the adjective dimensionless, and at the same time accompanied by the explanation, as for example with the case of dielectric permittivity, which certainly possesses dimensions in either metric systems (CGS or MKSA, or…), and its more convenient expression as relative dielectric permittivity, being the ratio with the dielectric constant of the vacuum as denominator.

Response 5: we modified it in the new revised manuscript, thank you. Actually in the ISEM2018 conference, there was a discussion about whether to call it real dielectric permittivity (since the complex dielectric permittivity consists of two parts: real and imaginary) or to call it relative dielectric permittivity (because it is a ratio of permittivity of medium to the permittivity of free space). Well, in the revised manuscript we follow your comment and we called it relative dielectric permittivity since it is more common in various studies (electronic, informatic and nature studies). Line20,28,53, 57-61,128, 144, 219, 220, 236,237, 242,243, 246,249,251,306

Line 57-63 for explanation of the model.

Thank you for your valuable comment.

Point 6: l. 61 I suggest being consistent with the units. In all figures, tables and the text you are using dS/m and not S.m-1 or S/m. I understand that these units were used by [3].

Response 6: we adjusted them to be described as dS/m, Thank you. Line57,61,139,191, Figure3,4,6

Point 7: l. 62 generic or general?

Response 7: it is “generic” as described by [3]

Point 8: l. 65 Apart from [15-17], perhaps another quite relevant work could be cited.( G. Kargas and P. Kerkides, 2010. Evaluation of a dielectric sensor for measurement of soil water electrical conductivity. Journal of Irrigation and Drainage Engineering (ASCE) 136 (8):553-558).

Response 8: we agree and added it in the revised manuscript. Thank you.

Point 9: l. 66 It is circuitry not circuity.  Also, apart from [8] the work of Kargas, George; Persson, Magnus; Kanelis, George; Markopoulou, Ioanna; Kerkides, Petros. Prediction of soil solution electrical conductivity by the permittivity corrected linear model using a dielectric sensor.  Journal of Irrigation and Drainage Engineering, Vol. 143, No. 8, 04017030, 01.08.2017 is quite relevant.

Response 9: Thank you, in the revised manuscript we correct the word and added Kargas et al 2017

Point 10: l. 87 heterogeneous and not heterogeneity.  Also, to modify and not modified…

Response 10: it is done.  Thank you

Materials and methods + Results and discussion

This section, to my opinion needs some further information and a more detailed description of the experiment and how this was performed. (Being an extension of a work presented in a Conference, this could be considered a good chance for the authors to feel free to develop and present their contribution without restrictions). For example,

Point 11: why did they use two columns and not more, or just one. Is this an insinuation of spatial variability?

Response 11:  We used two soil columns with the same condition to show that the offset of the Hilhorst model is not constant as he suggested for all moist soil or as other suggested that it is soil type depended [10,14,15, 16,] or soil type and salinity depended [17], we repeat the experiment to show that the offset changes even in the same soil type and the same conditions. But using Dynamic linear model enabled us to notice that the intercept changes and should be calculated before to estimate the pore water EC by Hilhorst model.  We make this clearer in result and discussion section, particularly, Line 117-120, 255-259

Point 12: Why the chosen flux (1L/h)

Response12: Thank you for your comment. We test different flux, and with 1l/h and 5 minutes’ interval between the observation of TDR data, we could depict the changes in the soil moisture for each depth. So our objective to collect the TDR data that could be applied for Hilhorst model in good visualization. In the revised manuscript Line 211-214

Point 13: and what was the hydraulic conductivity at saturation Ks of the column.

Response13: actually here we work on observation data. We got observation data and modeled them by statistical models.  We did not include in our model the soil properties, initial and boundary conditions to run our model.  now we are working on the numerical model (Hydrus 1D) where all these parameters are need to run the model.

Point 14: What is the purpose of the irrigation events, since the salinity level or the electrical conductivity of the moistening solution, which is the dominant factor, had only two different values, (20 and 30dS/m)

Response14: Thank you for your valuable comment. To apply dynamic linear model and kalman filter, time series of interested variable is needed, in our study, time series of εb , σb and εp are required to estimate σp. therefore, we used five irrigation events with two level of KCL solution to get variation of these interested variables over time.  In the revised manuscript we let this purpose clearer. Line148-152

Point 15: and in any case, how these values were compared with EC measured independently with the EC meter device, through the suction cups, or the values predicted by the Hilhorst model. Of course, in tables 2 and 3, as well as in fig.3 there is some information on the above. More explanation, on the findings, why for example, there are differences between the two columns, when these are moistened with equal EC KCl-solution and measurements refer to the same depth?

Response15: thank you for your valuable comment, in the new revised manuscript we explained well how we compare the EC estimated from the model with the EC meter, in brief, for each irrigation event we collect the one soil solution and get the measurement of EC by EC meter, we compared it with the mean of values of EC changes obtained from the model during the event. In the revised manuscript Line 195-199,269-263

Moreover, due the variability in water flow in unsaturated soil, we observed in our experiment a variation in the time needed to collect the sample, more time required to collect enough solution for EC meter more amount of ions gathered in the sample and as a result highe EC value. Therefore, there is difference in the EC values between the soil columns at the same depth (table 2). Line 199-203

Point 16:  The authors say something, concerning the feasibility of getting enough soil solution through the suction cups, but they do not report the number of these measured values and their comparison with the estimated ones.

Response16: we agree, as we mentioned above (response 15), in the revised manuscript we let this issue clear, Line 195-199

Point 17: In fig. 6, I would expect to see how these findings do compare with the measured σp values. Perhaps, some information from table 2 could be provided OR some more explanatory comments.

Response17: we agree, we put the values of σp from EC meter device in the figure 6.

Point 18: On the other hand, fig. 1 showing the conductance pathways seems not to be further commented in the text, thus, it appears to be redundant.

Response18: we agree, more information we added in the revised manuscript. Line 45-51

Point 19: The Gaussian white-noise errors wt and vt should be given below the lines of their associative equations and the symbols associated with the standard deviation of the normal probability distribution they are assumed to follow must not be given with the same symbol wt. It is confusing.

Response 19: we agree, in the last version there was a confusion. We changed them in the revised manuscript and let the variance with capital letter (we follow Petris, G. An R Package for Dynamic Linear Models. Journal of Statistical Software 2010, 36 (12).)

 Response 20: In table 2 where σp values are shown, are these values the mean of a number of measured values or what?

Response 20: we agree; in the previous version this is was not clear.  the values of EC in table 2 are the values of the samples measured by EC meter, as we mentioned above, we abled to collect one sample with enough solution for to measure by EC meter device for each irrigation event at each depth.

In the revised manuscript, we clear this explanation in the title of table 2 and in Line 195-199

Point 21: l. 98 What do you mean by the words “a two soil columns”

Response 21: we correct it, thank you Line 123

Point 22: l. 99 What do you mean by the words “by an irrigation head at …”

Response 22:  sorry for this confusion, we correct it Line 124

Point 23: l.139. Hilhorst

Response 23: we correct it, Thank you

Point 24: l. 192. The word “third” contradicts the legend of fig.4

Response 22: we correct it thank you so much

Reviewer 2 Report

General comment:

This paper deals with estimation of temporal change of soil pore water electrical conductivity by the use of dynamic linear model and Kalman filter. It seems interesting change to obtain more precise values for soil pore water electrical conductivity. It has merit to publish this paper in the journal. However, the manuscript is not enough discussions about the advantages of proposed method and disadvantages of conventional method. I still believe the calibration for specific soils with conventional method is useful to evaluate soil pore water content. This paper has just shown the proposed method. Discussions are not almost shown in the paper. Thus, the reader cannot judge the proposed method is useful of not. I have also raised some comments that the author should be addressed.

Specific comments:

1)L20: slop > slope?

2)L23: Kaman > Kalman?

3)L17-18: The author addressed "it still has not worked out very well.". Could you please add examples and references for this in the main body of the manuscript?

4)L62: It needs space between "4.1" and "as".

5)L64-65: The author addressed "it takes different values according to the soil type.". I could not understand why is it disadvantage for measurement. Calibration may work well.

6)L85: 5TE is not FDR sensor. It is capacitance probe.

7)L92-93: Disadvantages of deterministic model should be addressed. Please cite such kind of papers. Then, please explain about the weakness of that as written in the manuscript.

8)L98: What is the difference between these two columns?

9)L100, 102: 5 cm thickness of the layer was addressed two times. Please rephrase one of them.

10)L100: How did you control constant head of -30 hPa?

11)L106, Figure 2: Background dot-lines should be removed.

12)L108: Does cup mean porous-cup?

13)L111: Does irrigation-head mean pressure-head? This may confuse readers. In this figure constant pressure head cannot be applied on the soil surface. How much did you applied flux on the soil surface? Exact values should be addressed.

14)L143: Package'dlm' Please shown exact name.

15)L149: Character of the offset is very different in between equation and main text of the manuscript. Please use same character for the same variable.

16)L151: How about column 1? I still could not understand why did the author use two same conditions columns in the experiment.

17)L152: slop > slope?

18)L163 (same as L169, L190): Expressions should be improved. It is not acceptable to begin each paragraph with "Figure 1 show" or "Table 1 shows". I suggest that the authors take a careful look at the structure of published papers.

19)L191: What about soil column 1?

20)Theory presented in this paper is as same as the author published one (21). The author has already applied this method on site. Why is it in need to test on the column experiment? It should be more clarified.

Author Response

Dear Editor and Referees,

Thank you for your very careful review of our manuscript, and for the comments, corrections that ensued. A major revision of the manuscript has been carried out to take all of them into account. this this cover letter, we believe that the manuscript has been significantly improved.

in the present letter, we detail the major changes in the manuscript to address the issues raised by the review

#Response to Reviewer 2 Comments

General comment:

This paper deals with estimation of temporal change of soil pore water electrical conductivity by the use of dynamic linear model and Kalman filter. It seems interesting change to obtain more precise values for soil pore water electrical conductivity. It has merit to publish this paper in the journal.

Thank you for your comments

Point: However, the manuscript is not enough discussions about the advantages of proposed method and disadvantages of conventional method. I still believe the calibration for specific soils with conventional method is useful to evaluate soil pore water content.

Response: We agree with you that the conventional methods (Extracting soil solution by suction or using saturated paste conductivity measurements) are useful and effective, we mentioned about their disadvantage regarding labour-intensive.  And in the new revised we explain more about the advantage of our proposed method Line 115-120 and 255-269

Specific comments:

Point1: This paper has just shown the proposed method. Discussions are not almost shown in the paper. Thus, the reader cannot judge the proposed method is useful of not.

Response1: In the results and discussion section we let this issue clearer. thank you so much for your comment

 I have also raised some comments that the author should be addressed.

Specific comments:

Point 2: L20: slop > slope?

Response 2: Thank you so much, we correct it in the revised manuscript

Point 3: L23: Kaman > Kalman?

Response 3: we correct it, Thank you

Point 4: L17-18: The author addressed "it still has not worked out very well.". Could you please add examples and references for this in the main body of the manuscript?

Response 4: thank you for your comments, you are right that we did not let this information with reference, actually many studies worked and still work on the soil electromagnetic sensors data and used modeds and methods to estimate pore water EC in situ. in our work we mentioned about studies that used just Hilhorst model to estimate pore water EC. So, they concluded with different recombination to use the model offset.  Moreover, we get this conclusion during the discussion with Prof. Gaylon Camppll,(Prof. Campell published a slim volume entitled soil Physics with Basic, Campbell, 1985. this text book was one of the first and best publication to show the potential for numerical computer models to solve applied problems in the field of soil physics. He still works in his company METER and I still contact his group there regarding the developing of soil salinity sensor.)

In the revised manuscript I mentioned about this information Line89-90

Thank you for your comment

Point 5: L62: It needs space between "4.1" and "as".

Response 5: Thank you, we correct it

Point 6: L64-65: The author addressed "it takes different values according to the soil type.". I could not understand why is it disadvantage for measurement. Calibration may work well.

Response 6: we agree and actually this is the point, calibration will work well, but for example the Delta sensor uses Hilhorst model and they insert in their software the offset to be 4.1 for all soils and this does not work as we mentioned in our work. So our model takes this point into account and calculated the offset according to the site conditions.

in the revised manuscript we make this point clearer, Line 72-79, 115-120, 240-248, 264-269

Thank you for your comment.

Point 7: L85: 5TE is not FDR sensor. It is capacitance probe.

Response 7: it is FDR or capacitance probe. The capacitance and FDR are the same , they  determines the dielectric permittivity of a medium by measuring the charge time of a capacitor, which uses that medium as a dielectric. Site from the manufacture explains this  http://www.environmentalbiophysics.org/tdr-versus-capacitance-or-fdr/ and also from their site  http://ictinternational.com/products/5te/decagon-5te-vwc-temp-ec/

Point 8: L92-93: Disadvantages of deterministic model should be addressed. Please cite such kind of papers. Then, please explain about the weakness of that as written in the manuscript.

Response 8: Thank you for your comment. in the revised manuscript we explained better our idea. Line 81-99

Point 9: L98: What is the difference between these two columns?

Response 9: thank you for your comment. in the the revised manuscript we explain better why we use two columns with the same conditions. In brief, we used two soil columns with the same condition to show that the offset of the Hilhorst model is not constant as he suggested for all moist soil or as other suggested that it is soil type depended [10,14,15, 16,] or soil type and salinity depended [17], we repeat the experiment to show that the offset changes even in the same soil type and the same conditions. But using Dynamic linear model enabled us to notice that the intercept changes and should be calculated before to estimate the pore water EC by Hilhorst model.  We make this clearer in result and discussion section, particularly, Line 117-120, 255-259

Point 10: L100, 102: 5 cm thickness of the layer was addressed two times. Please rephrase one of them.

Response 10: we rephrase it, thank you

Point 11: L100: How did you control constant head of -30 hPa?

Response 11: We used the vacuum pump. we make this clear in the revised manuscript Line 124

Point 12: L106, Figure 2: Background dot-lines should be removed.

Response 12: Thank you we, in the revised manuscript, we removed it

Point 13: L108: Does cup mean porous-cup?

Response 13: Yes, we men porous suction cups, we let this clear in the revised manuscript, thank you, Line 133,192,195, 262

Point 14: L111: Does irrigation-head mean pressure-head? This may confuse readers. In this figure constant pressure head cannot be applied on the soil surface. How much did you applied flux on the soil surface? Exact values should be addressed.

Response 14: we agree, there is some confusion in our explanation. We meant by head as “sprinkler” so we correct this error in the revised manuscript Line 124

Point 15: L143: Package'dlm' Please shown exact name.

Response 15: its package dlm , as described form his author (Petris, G. An R Package for Dynamic Linear Models. Journal of Statistical Software 2010, 36) Line 173-175

Point 16: L149: Character of the offset is very different in between equation and main text of the manuscript. Please use same character for the same variable.

Response 16: Thank you so much we correct it in the revised manuscript

Point 17: L151: How about column 1? I still could not understand why did the author use two same conditions columns in the experiment.

Response 17: Thank you for your comment, we agree that there is some confution about why we worked in two soil column with same conditions.

In the revised manuscript we explain better our purpose, please see our answer in Point 8

Point 18: L152: slop > slope?

Response 18: Thank you, we correct it

Point 19: L163 (same as L169, L190): Expressions should be improved. It is not acceptable to begin each paragraph with "Figure 1 show" or "Table 1 shows". I suggest that the authors take a careful look at the structure of published papers.

Response 19: We did it and we improve the writing in the revised manuscript, thank you so much

Point 20: L191: What about soil column 1?

Response 20: we just want to show the plot of σb, εb and εp variables, so we choose column 2 as an example, we let this clear in the revised manuscript Line243

Point 21: Theory presented in this paper is as same as the author published one (21). The author has already applied this method on site. Why is it in need to test on the column experiment? It should be more clarified.

Response 21: we agree, in the revised manuscript we explain better the purpose of work and its new contribution from previous work, Line 113-120, 2255-269, 316-317

Reviewer 3 Report

In my opinion, the paper is well written and contributes to the existing knowledge. I could not find any logical errors in the presentation, and the approaches used. In order to improve the manuscript, find my remarks and comments below.

You need to clarify two issues

i)                    Why the values of the bulk EC in Figure 3 are much larger than those shown in Figure 4

ii)                  If the values of the bulk electrical conductivity in Figure 3 are correct then you should consider whether the relationship σb - εb is linear

l.61. I think that this value (0.3 S/m) refers to bulk electrical conductivity 

l.65-66 You can also include the following papers

G Kargas, M Persson, G Kanelis, I Markopoulou, P Kerkides. 2017. Prediction of soil solution electrical conductivity by the permittivity corrected linear model using a dielectric sensor. Journal of Irrigation and Drainage Engineering 143 (8), 04017030.

G. Kargas and P. Kerkides, 2010. Evaluation of a dielectric sensor for measurement of soil water electrical conductivity. Journal of Irrigation and Drainage Engineering (ASCE) 136 (8):553-558.

l.98. Why are two columns used?

l.101 Since the column is homogeneous (bulk density 1.4 g/cm3) , why do you report the 11 layers

l.110 and 111. You should use minus before the values

Figure 6. How to explain the smallest offset values at the smallest depth (21 cm). Any idea?

Author Response

Dear Editor and Referees,

Thank you for your very careful review of our manuscript, and for the comments, corrections that ensued. A major revision of the manuscript has been carried out to take all of them into account. this this cover letter, we believe that the manuscript has been significantly improved.

in the present letter, we detail the major changes in the manuscript to address the issues raised by the review

#Response to Reviewer 3 Comments

Comments and Suggestions for Authors

In my opinion, the paper is well written and contributes to the existing knowledge. I could not find any logical errors in the presentation, and the approaches used. In order to improve the manuscript, find my remarks and comments below.

Thank you for your comments

You need to clarify two issues

Point 1: Why the values of the bulk EC in Figure 3 are much larger than those shown in Figure 4

Response 1: you are right, there was different in the scale, in the revised manuscript we correct it. thank you for your comment

Point 2:If the values of the bulk electrical conductivity in Figure 3 are correct then you should consider whether the relationship σb - εb is linear

Response 2: Thank you, we correct the figure 3

Point 3l.61. I think that this value (0.3 S/m) refers to bulk electrical conductivity 

Response3: Please see Hihorst work [3] it is pore water electrical conductivity

Point 4:l.65-66 You can also include the following papers

G Kargas, M Persson, G Kanelis, I Markopoulou, P Kerkides. 2017. Prediction of soil solution electrical conductivity by the permittivity corrected linear model using a dielectric sensor. Journal of Irrigation and Drainage Engineering 143 (8), 04017030.

 G. Kargas and P. Kerkides, 2010. Evaluation of a dielectric sensor for measurement of soil water electrical conductivity. Journal of Irrigation and Drainage Engineering (ASCE) 136 (8):553-558.

Response 4: Thank you for your recommendation, they are very related and we include them.

Point 5: l.98. Why are two columns used?

Response 5: Thank you for your valuable comment. Please see our answer to reviewer 1 (Point 11) and reviewer 2 (Point 8)

Point 6: l.101 Since the column is homogeneous (bulk density 1.4 g/cm3) , why do you report the 11 layers

Response 6: thank you, we just want to show the way of construction, but you are right by making some confusion. now in the revised manuscript we correct it Line125

Point 7: l.110 and 111. You should use minus before the values

Response 7: we agree; in the revised manuscript we correct it Line 135-136

Point 8: Figure 6. How to explain the smallest offset values at the smallest depth (21 cm). Any idea?

Response 8: actually there is a lot of factor could effect on the offset variation along the depth, soil temperature could be one reason but unfortunately in our experiment the temperature did not change a lot along the soil column to see its effect. This could be achieved by irrigate with different water temperature and see if it has an effect on the offset variation. Thank you so much for your comment and this could an idea  be for our future studies.

Round  2

Reviewer 1 Report

The revised MS has been improved and in this respect it warrants publication. There are some minor spelling or syntactical errors, such as:

l.32 decreases......and causes...

l.85 deterministic models (not model)

l.92-95 To become clearer.

l.264 filter not fitler.

l.265 Hilhorst model (not mdoel).

l.265 two not tow.

Still fig.1 originally was proposed by Sauer et al. Industrial and Engineering Chemistry, 1955, pp.2187-2193 and it should be an honor for the authors to find a way to mention it. I am aware that this will be a problem since the numbers associated with the list of references will have to be changed. This way of dealing with citations and references (Numbers and not names, or occasionally names), to my opinion is wrong. I believe that it is a kind of respect to refer to an achievement, or even an idea, to the people (using directly their names and not sparing space), they proposed that idea or made that achievement. I should be very happy if the Editors of MDPI change the current way of references (mainly in the text).

Author Response

Dear Reviewer,

Thank you for your very careful review of our manuscript. in the present letter we detial the changes in the manuscirpt to address the issues reaised by your commets.

The revised MS has been improved and in this respect it warrants publication. There are some minor spelling or syntactical errors, such as:

Point 1: l.32 decreases......and causes...

Response 1: we correct it.  Thank you

Point 2: l.85 deterministic models (not model)

Response 2: we correct it. Thank you

Point 3: l.92-95 To become clearer.

Response 3: we rephrase it. Thank you

Point 4: l.264 filter not fitler.

Response 4: we correct it. Thank you

Point 5: l.265 Hilhorst model (not mdoel).

Response 5: we correct it. Thank you

Point 6: l.265 two not tow.

Response 6: we correct it. Thank you

Point 7: Still fig.1 originally was proposed by Sauer et al. Industrial and Engineering Chemistry, 1955, pp.2187-2193 and it should be an honor for the authors to find a way to mention it. I am aware that this will be a problem since the numbers associated with the list of references will have to be changed. This way of dealing with citations and references (Numbers and not names, or occasionally names), to my opinion is wrong. I believe that it is a kind of respect to refer to an achievement, or even an idea, to the people (using directly their names and not sparing space), they proposed that idea or made that achievement. I should be very happy if the Editors of MDPI change the current way of references (mainly in the text).

Response 7:  Actually, we follow the literature review where they mentioned to Rhoades et al (1989), but when I read the paper you mentioned I noticed that the suggestion of three conductance pathways to estimate bulk electrical conductivity is done by others and before Rhoades et al (1989) work. So in the discussion part of Sauer et al (1955) they mentioned that the three conductance pathways are suggested first by Wyllie and Southwick  (1954) and I found that the figure 6 of Wyllie and Southwick (1954) explains the same way of these pathways.   We agree totally with you to mention about the first people who suggested this way to estimate the bulk electrical conductivity. In the new version of our work you could find his in the figure 1 and in the text Line 46. thank you for your comment.

Reviewer 2 Report

Author addressed most of the problems raised by reviewer. I have raised some minor problems again that should be addressed by author.

1)    Related to no.9 (1st round review), I think authors response seems more understandable than revisions added in the revised manuscript. Is it possible to use this explanation in the revised manuscript?

2)    Related to no.19 (1st round review), The author did not understand my suggestion. The author still used “Figure 3 shows (Line 217)”, “Figure 4 shows (Line 249)” in the revised manuscript. It is not acceptable to begin with each paragraph with these. Please revise these expressions.

3)    Line 127, Period between m3 and m-3 should be removed. However, to unify the expression of the units, how about to use m3/m3? Because the author used “/” for other units.

4)    Line 45-55, Width of the text seems narrow.

5)    Line 72-79, Could you please rephrase this part because “offset=4.1” are repeated many times.

6)    Line 189, Table 1, In “1/σp”, p should be in subscript.

7)    Line 197, Table 2, “cm” could not split into two lines.

8)    Line 216, Table 3, What is the maximum number of significant figures? It seems too much, seven significant figures. I think three to four figures may be enough for this case.

9)    Line 259, In “σp”, p should be in subscript.

10) Line 271, what is “tow”?

Author Response

Dear Reviewer,

Thank you for your very careful review of our manuscript. in the present letter we detial the changes in the manuscirpt to address the issues reaised by your commets.

Point 1:   Related to no.9 (1st round review), I think authors response seems more understandable than revisions added in the revised manuscript. Is it possible to use this explanation in the revised manuscript?

Response 1: Thank you, we take your comment and add the justification why “we two columns” in the conclusion part, Line 315-321.

Point 2:     Related to no.19 (1st round review), The author did not understand my suggestion. The author still used “Figure 3 shows (Line 217)”, “Figure 4 shows (Line 249)” in the revised manuscript. It is not acceptable to begin with each paragraph with these. Please revise these expressions.

Response 2: we correct it. Thank you, Line …………..

Point 3:     Line 127, Period between m3 and m-3 should be removed. However, to unify the expression of the units, how about to use m3/m3? Because the author used “/” for other units.

Response 3: We rewrite it. Thank you

Point 4:     Line 45-55, Width of the text seems narrow.

Response 4: We adjust it. Thank you

Point 5:    Line 72-79, Could you please rephrase this part because “offset=4.1” are repeated many times.

Response 5: we rephrase it. Thank you Line………..

Point 6: Line 189, Table 1, In “1/σp”, p should be in subscript.

Response 6: we correct it. Thank you

Point 7: Line 197, Table 2, “cm” could not split into two lines.

Response 7: we adjust it. Thank you

Point 8: Line 216, Table 3, What is the maximum number of significant figures? It seems too much, seven significant figures. I think three to four figures may be enough for this case.

Response 8: you are right. Three number is enough, we adjust them.

Point 9: Line 259, In “σp”, p should be in subscript.

Response 10: we adjust it. Thank you

Point 10: Line 271, what is “tow”?

Response 10: it is “two”, we correct it. Than you